# Rapid Sample Screening Method for Authenticity Controlling of Vanilla Flavours Using Liquid Chromatography with Electrochemical Detection Using Aluminium-Doped Zirconia Nanoparticles-Modified Electrode

**DOI:** 10.3390/molecules27092915

**Published:** 2022-05-03

**Authors:** Yassine Benmassaoud, Khaled Murtada, Rachid Salghi, Mohammed Zougagh, Ángel Ríos

**Affiliations:** 1Department of Analytical Chemistry and Food Technology, Faculty of Pharmacy, University of Castilla-La Mancha, 13071 Ciudad Real, Spain; yassinebenmassaoud@gmail.com (Y.B.); kmurtada@uwaterloo.ca (K.M.); mohammed.zougagh@uclm.es (M.Z.); 2Analytical-NANO-Group, Regional Institute for Applied Chemistry Research (IRICA), 13071 Ciudad Real, Spain; 3Laboratory of Applied Chemistry and Environment, Ecole Nationale Des Sciences Appliquées (ENSA), Université Ibn Zohr, P.O. Box 1136, Agadir 80000, Morocco; r.salghi@uiz.ac.ma

**Keywords:** aluminium doped zirconia nanoparticles, screen-printed carbon electrode, HPLC-electrochemical detection, vanilla, food samples

## Abstract

A rapid and sensitive technique for frauds determination in vanilla flavors was developed. The method comprises separation by liquid chromatography followed by an electrochemical detection using a homemade screen-printed carbon electrode modified with aluminium-doped zirconia nanoparticles (Al-ZrO_2_-NPs/SPCE). The prepared nanomaterials (Al-ZrO_2_-NPs) were characterized by using X-ray diffraction (XRD), transmission electron microscopy (TEM) and energy dispersive X-ray (EDX). This method allows for the determination of six phenolic compounds of vanilla flavors, namely, vanillin, p-hydroxybenzoic acid, p-hydroxybenzaldehyde, vanillyl alcohol, vanillic acid and ethyl vanillin in a linear range between 0.5 and 25 µg g^−1^, with relative standard deviation values from 2.89 to 4.76%. Meanwhile, the limits of detection and quantification were in the range of 0.10 to 0.14 µg g^−1^ and 0.33 to 0.48 µg g^−1^, respectively. In addition, the Al-ZrO_2_-NPs/SPCE method displayed a good reproducibility, high sensitivity, and good selectivity towards the determination of the vanilla phenolic compounds, making it suitable for the determination of vanilla phenolic compounds in vanilla real extracts products.

## 1. Introduction

Nowadays, mixed food products are becoming very popular in the food industry [1], and their consumption in Europe has significantly increased. Thus, the quality control in this area is crucial to avoid the occurrence of any side effects. Vanilla is a type of tropical climbing plant [2] that is considered a major flavouring product, finding its application in many fields, namely food, cosmetics and pharmaceuticals. It is naturally obtained from vanilla plantifolia extract and includes vanillin (Van) as the major constituent, together with several related compounds such as p-hydroxybenzaldehyde (p-HB), p-hydroxybenzoic acid (p-HBA), vanillic acid (VA) and vanillyl alcohol (V-OH) [3]. Given its popularity, the high demand and cost of natural vanilla give rise to many adulterations carried out using Ethyl vanillin (EVan) or synthetic Vanillin.

Until now, several analytical methods have been used for the determination of the chemical constituents of vanilla flavour, including gas and liquid chromatography (GC & LC) [4,5,6] and capillary electrophoresis (CE) techniques [7]. Electrochemical detection (ECD) methods are also used for the determination/detection of vanilla phenolic compounds because of their electro-active properties. ECD is an important technique for the determination of vanilla phenolic compounds due to its simplicity, fast response, high sensitivity, and cheap instrumentation [2,8,9,10]. For the time being, several analytical methods have been used for the determination/detection of frauds in vanillin products, including CE [7], CE microchip approach with ECD [11,12], and liquid chromatography-mass spectrometry (LC-MS) [5]. Each one of these methods has drawbacks for the detection/determination of the fraud in the vanilla product. Based on these publications, a new method was developed for rapid screening for the authenticity controlling of vanilla flavours using LC and ECD.

A various number of newly developed electrochemical sensors were recently reported. Screen printed carbon electrodes (SPCEs) were specially used in many fields including food, environment and biomedicine [2,13]. In fact, SPCEs are inexpensive and could be utilised with a small sample volume as a droplet in a miniaturised system. Nevertheless, the over-potential in the oxidation of vanillin and the other related compounds on several unmodified electrode surfaces is significant. The intervention of nanomaterials as electrode surface modifiers provides both a better selectivity and sensitivity [14,15]. To remediate this, several modified electrodes were tested, such as graphene [16], aluminum-doped titanium oxide nanoparticles (Al-TiO_2_-NPs) [2], aluminium-doped copper selenide nanoparticles (Al-CuSe-NPs) [17], silver nanoparticles (Ag NPs) [18], gold nanoparticles (Au NPs) [19], reduced graphene oxide with copper selenide (CuSe@rGO) [15] and multi-walled carbon nanotubes (MWCNTs) [20], molecularly imprinted polymer (MIP) [21], and others [22].

In this work, a new screening electrochemical method for the control of vanilla flavours is developed using aluminium-doped zirconia nanoparticles as a modifying film for a screen-printed carbon electrode (Al-ZrO_2_-NPs/SPCE). A comparison was then carried out, proving that the homemade sensor provided a better sensitivity than both a non-modified SPCE and carbon nanotubes-modified screen-printed carbon electrode (CNT-SPCE) while being applied to phenolic vanilla compounds detection. The performances of the modified electrode (Al-ZrO_2_-NPs/SPCE) were then used for the discrimination of frauds in vanilla flavours and applied to numerous food samples.

## 2. Results

### 2.1. Characterisation of Aluminium-Doped ZrO_2_ Nanoparticles (Al-ZrO_2_-NPs)

The XRD patterns of Al nanoparticles and Al-ZrO_2_-NPs illustrated in Appendix A show the predominance of Al-NPs and represent the face-centered cubic (fcc) form of aluminium. Otherwise, the Al-ZrO_2_-NPs is shown to be constituted from Al-NPs as a major compound together with ZrO_2_-NPs with less presence. Using the Scherrer equation (D = Kλ/βcosθ), it was also concluded that the average particle sizes varied for the films as Al (7.63 nm) and Al-ZrO_2_ (12.75 nm). TEM and EDX micrographs for the Al-NPs and Al-ZrO_2_-NPs are shown in Appendix A. The EDX micrographs show the surface morphology difference between the different materials. Appendix A shows the TEM and EDX micrographs of the Al-NPs without doping with ZrO_2_-NPs. A large number of precipitates were distributed homogeneously in the aluminium grains. The homogenous distribution of the Al-NPs leads to an increase in the hardness and tensile properties. Appendix A shows the TEM and EDX micrograph of the Al-ZrO_2_-NPs; the ZrO_2_-NPs were homogeneously distributed through the Al grains, indicating that a large amount of Al-NPs had been doped in the ZrO_2_-NPs. The TEM and EDX micrograph results confirm the XRD results. 

### 2.2. Optimisation the Experimental Parameters 

The detection conditions of the vanilla relative compounds were optimised, starting with the Al-ZrO_2_-NPs modification volume. After testing 2.0, 4.0 and 6.0 µL, the optimum volume was 2.0 µL. Then, a pH condition study showed that the optimum results were reached under pH 7.4, after testing a pH range from 1 to 12.

For the chromatographic analysis, different mobile phase compositions of water containing 0.1% (formic acid, acetic acid and phosphoric acid) were tested, and the best results were accomplished when water/phosphoric acid (0.1% *v*/*v*) was used. Methanol, ethanol and acetonitrile were tested as organic solvents, and due to its compatibility with the mobile phase, acetonitrile was selected.

The flow rate effect was then studied between 0.5–1.0 mL min^−1^, and 1.0 mL min^−1^ provided the best compromises between the retention times and peak area. Moreover, the injection volume was studied within the 10–40 µL range, demonstrating that a 40 µL aliquot provided optimum retention times and peak area.

Different electrodes were then tested and illustrated in Figure 1, which shows the comparison of LC chromatograms obtained at non-modified SPCE, CNT-SPCE and Al-ZrO_2_-NPs/SPCE for 40 µL of 25 mg L^−1^ solutions of Van, Evan, p-HB, p-HBA, V-OH and VA injected in the mobile phase H_2_O:H_3_PO_4_ (0.1%, *v*/*v*) at pH 7.4 and 1.0 mL min^−1^ flow rate. As can be seen, the peak currents are considerably larger for all vanilla phenolic compounds at Al-ZrO_2_-NPs-SPCE than at CNT-SPCE and bare SPCE. The performance of the new sensor was also evaluated by a cyclic voltammetry comparison with a non-modified electrode, as shown in (Appendix A).

The choice of the applied potential to Al-ZrO_2_-NPs/SPCE as an amperometric detector was carried out by plotting the measured current values at different applied potentials within the range of 0.3–1.1 V after the injection of a 40 µL aliquot of 25 µg mL^−1^ Van, Evan, p-HBA, p-HB, VA and VOH solution. The maximum current was obtained at +0.9 V for the six phenolic compounds (Figure 2). Under the previous conditions, only conditioning the prepared electrochemical sensor at the beginning of every experiment was enough, and no cleaning or pretreatment step was needed. Moreover, the modified electrode was regenerated every 20 analyses, and the modifier dispersion was prepared every week to maintain the same electrochemical detection and to avoid any type of contamination.

### 2.3. Validation of the Methodology

Under the optimised conditions, concentrations ranging between 0.5–25 µg g^−1^ of vanilla phenolic compounds standard solution were injected, separated and determined. As shown in Figure 3, the analysis was achieved in less than 11 min, providing a high resolution. The precision of the present method was expressed by relative standard deviation (RSD%) for each one of the vanilla relative compounds. The intercept, the regression coefficient of the respective calibration curves as well as the theoretical limits of detection (LODs) and limits of quantification (LOQs) were also calculated (Table 1).

### 2.4. Analytical Application 

The efficiency and reliability of the proposed method was tested by looking into the origin of the natural vanilla extract samples (A)–(D), as demonstrated in Table 2. First, the identification of Van together with secondary markers (p-HBA, p-HB and VA) and the absence of Evan authenticate the natural origin of the sample. This case was illustrated by the phenolic profile of sample (A) (Figure 4A). The second case, showed by the detection of Evan in both samples (B) and (D) (Figure 4B,D), lead to the conclusion of the non-natural origins of the vanilla extract products. 

The analysis of the last real vanilla extract product (C) (Figure 4C) presented a complicated case. In fact, only Van was detected, without the related phenolic compounds (p-HBA, p-HB and VA). This absence could be confirmed, leading to the conclusion that the sample was from a non-natural origin or that perhaps the lack of sensitivity of the present method led to its absence. To elucidate this last situation, a study of the concentrations of secondary markers, associated with the Van concentration found in sample (C), was carried out and evaluated by a comparison with typical ratio values (p-HB/Van, p-HBA/Van and VA/Van) of natural vanilla extracts [5,12,23,24,25]. As reported in Table 3, the expected concentration of p-HB ranged between 0.51–1.02 µg g^−1^, providing an interval higher than the LOQ of the marker. Thus, the real absence of p-HB can be concluded. On the other hand, the LOQs were respectively lower and within the expected concentration ranges for p-HBA and VA (Table 3). Despite this, the non-natural providence of sample C could be concluded owning to the absence of p-HB. The developed method provides clear and good advantages in terms of method sensitivity when compared to previously published reports on the electrochemical determination of vanillin involving the use of other vanillin sensors and electrochemical detections (Table 4).

## 3. Materials and Methods

### 3.1. Materials and Standards

Aluminium acetylacetonate (Al(acac)_3_, 99%), lithium tetrahydruroaluminate (LiAlH4, 95%), mesitylene (97%), polyacrilyc acid, hexafluorozirconic acid, EVAN (ethyl vanillin 98%), V-OH (vanillic alcohol 98%), p-HB (p-hydroxybenzaldehyde 98%) were purchased from Sigma-Aldrich (St. Louis, MO, USA), while Nafion 117 solution (5% in a mixture of lower aliphatic alcohols and water), VAN (vanillin), p-HBA (p-hydroxybenzoic Acid) and VA (vanillic acid) were purchased from Fulka chemie (Buchs, UK). Hydrofluoric and phosphoric acid (85%) were obtained from Panreac Applichem (Barcelona, Spain). Methanol, acetonitrile and ethanol (HPLC Grade) were purchased from Fisher Chemical (Hampton, NJ, USA). Vanilla real extract products (A–D) were purchased from local markets (Ciudad Real, Spain). High-quality water purified in a Milli-Q system was used (18.2 MΩcm (25 °C) and a TOC value below 5 ppb). 

All stock solutions were dissolved in ethanol to obtain a final concentration of 0.1 M and were preserved at 4 °C. The working solutions were made by dilution in the LC mobile phase solution to the desired concentrations. 

### 3.2. Instruments and Apparatus

The liquid chromatography system used for all determinations was an Agilent 1200 composed of an LC pump, an auto sampler (40-µL injection loop), a degasser and a thermostat column compartment. A reversed-phase Luna 5 µm PFP (2) 100 A (150 × 4.6 mm) analytical column was used for the chromatographic separation (no pre-column was used). The system was coupled to a CHI812D electrochemical analyzer with an amperometric detection from CH Instruments (Austin, TX, USA), and the data was processed using CHI812D electrochemical software. Screen-printed carbon electrodes (SPCE) provided by Dropsens were utilised for all the measurements. A philips model X’Pert MPD diffractometer using a CuKα source (λ = 1.5418 Å), programmable divergence slit, graphite monochromator and proportional sealed xenon gas detector was used under 40 KV, intensity 40 mA and an angular range of 20 to 70 degrees (2 Theta), a step of 0.02 degrees 2 Theta and a time per step of 1.50 s. A Jeol JEM 2011 operating at 200 kV and equipped with an Orius Digital Camera (2 × 2 MPi) was used for TEM measurements. A Digital Micrograph TM 1.80.70 for GMS 1.8.0 Gatan was employed for HRTEM analysis. A lacey carbon/format-coated copper grid was used as a support for the sample preparation by deposition of a drop of the synthesised material suspension. A basic 20 pH-meter (Crison Instruments S.A, Alella, Spain) and an ultrasound bath (Selecta, Barcelona, Spain) were also used.

### 3.3. Preparation of Aluminum-Doped ZrO_2_ Nanoparticles (Al-ZrO_2_-NPs)

At first, Al(acac)_3_ was used for the preparation of Al-NPs by mixing the Al(acac)_3_ with LiAlH4 and refluxing at 165 °C for about 72 h. To prepare Al-NPs, LiAlH_4_ (30 mmol) was added to a solution of Al(acac)_3_ (10 mmol) and Mesitylene already mixed in a three-necked round-bottom flask. The reaction was refluxed, meanwhile purged with N_2_ and stirred for 72 h at 165 °C. Then, the mixture was cooled down to room temperature. The formed precipitate was then crushed and vacuum-dried for 3–4 h. Using 25 mL of cold methanol, the crude product was washed three times in the aim of avoiding any exothermic reaction between the solvent and Al-NPs. The excess starting materials were washed with methanol, filtered, and finally dried under low pressure. Polyacrylic acid (12.06% *w*/*w*) was taken in (87.9% *w*/*w*) of pure water, (0.04% *w*/*w*) of hydrofluoric acid was added, and the mixture was mixed for 15 min. In a separate beaker, hexafluorozirconic acid solution (83.7% *w*/*w*) was taken in (83.7% *w*/*w*) of pure water and mixed for 10 min. In another separate beaker, (1.2%) of the first bath and (1.7%) of the second bath were taken. Al-NPs (0.5 g) were taken in a zirconium bath (25 mL). The mixture was mixed for 3 h, filtrated and dried.

### 3.4. Preparation of Modified Electrode 

A concentration of 1 mg/mL was prepared by dispersing Al-ZrO_2_-NPs in water (0.5% Nafion, *v*:*v*) [2,17]. Then, the modified SPCEs were prepared by dropping 2 μL of the previously dispersed Al-ZrO_2_-NPs on the electrode surface. The modified sensor was then dried under infrared light for 20 min. The electro-active surface area of the non-modified electrode was compared with the new sensor by a cyclic voltammetry run of 0.1 mM of dopamine in a phosphoric acid solution (0.1 M) as electrolyte (Appendix A). 

### 3.5. Sample Preparation

The real samples (A, B, C, and D) were prepared by accurately weighting and dissolving 25 mg in 25 mL of ethanol using a volumetric flask and were preserved at 4 °C. All prepared samples were protected from the light, filtered with a 0.45 mm nylon filter (Millipore Millex-HN) prior to their use and used within 24 h after preparation.

### 3.6. Chromatographic Conditions and Amperometric Detection at Al-ZrO_2_-NPs/SPCE

Anterior LC methods used C_18_ columns with distinct characteristics and different mobile phase compositions for the separation of vanilla phenolic compounds [5]. A mobile phase consisting of (A) water: phosphoric acid (0.1% *v*/*v*) and (B) acetonitrile with the following gradient: 20% B for 7 min, 40% B until 14 min and 20% B for one minute was used. A flowrate of 1.0 mL/min and an injection volume of 40 μL were chosen. The column temperature was maintained at 25 °C. The applied potential for the amperometric detection was +0.90 V for both standard real sample solutions. A post-run time of 20 min was performed for the re-equilibration of the column. 0.45 μm nylon membranes were used to filtrate the solvents before their use.

## 4. Conclusions

In this work, a new analytical method was developed for spotting frauds in vanilla products by the selective detection of vanillin and five other related compounds. This method used liquid chromatography with amperometric detection on a newly modified screen-printed carbon electrode (Al-ZrO_2_-NPs/SPCE). A simple, effective, and rapid electrochemical detection approach was proposed using a home-made sensor for the control of adulterations in vanilla food samples. By increasing the sensitivity and providing a higher stability and selectivity, vanilla together with five other related compounds were successfully discriminated in order to distinguish the natural origins from the non-natural ones. Nevertheless, some limitations of the use of Al-ZrO_2_-NPs/SPCE involve the short electrode lifetime, poor inter-electrode reproducibility, and the fact that they are not user-friendly.

## Figures and Tables

**Figure 1 molecules-27-02915-f001:**
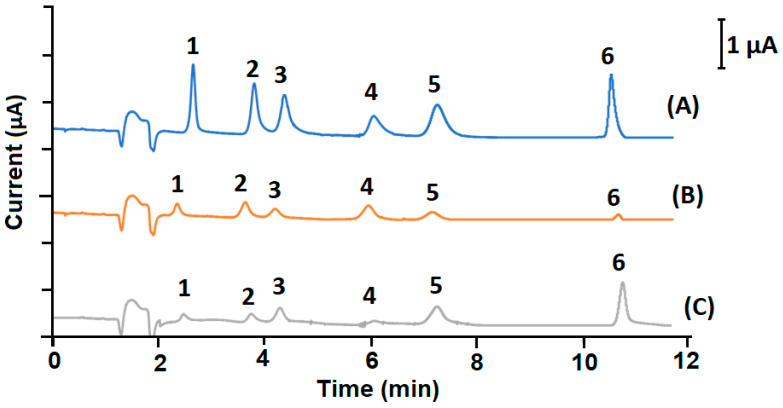
Comparison between the electrochemical response of vanilla phenolic compounds at (**A**) Al-ZrO_2_-NPs/SPCE, (**B**) non-modified SPCE and (**C**) CNT-SPCE. Peak identification; 1: V-OH, 2: p-HBA, 3: VA, 4: p-HB, 5: Van and 6: Evan.

**Figure 2 molecules-27-02915-f002:**
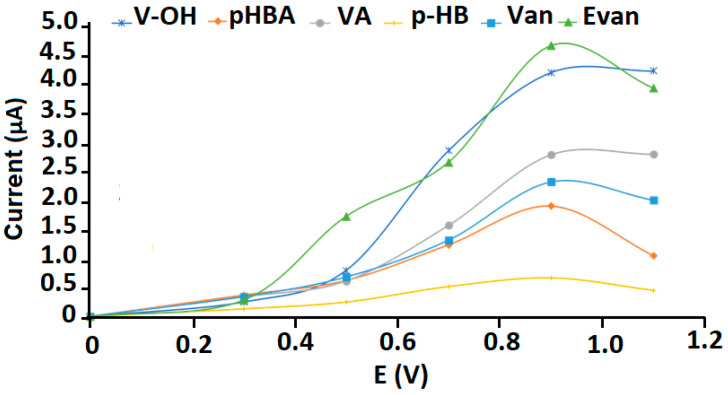
Hydrodynamic voltammogram showing the impact of the applied potential to the Al-ZrO_2_-NPs-modified SPCE on the analytical signal of the six vanilla phenolic compounds.

**Figure 3 molecules-27-02915-f003:**
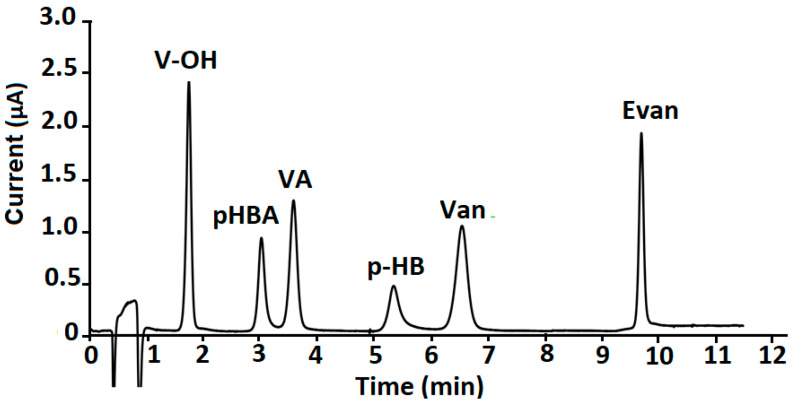
Chromatogram of the six vanilla phenolic compounds with amperometric detection at Al-ZrO_2_-NPs/SPCE.

**Figure 4 molecules-27-02915-f004:**
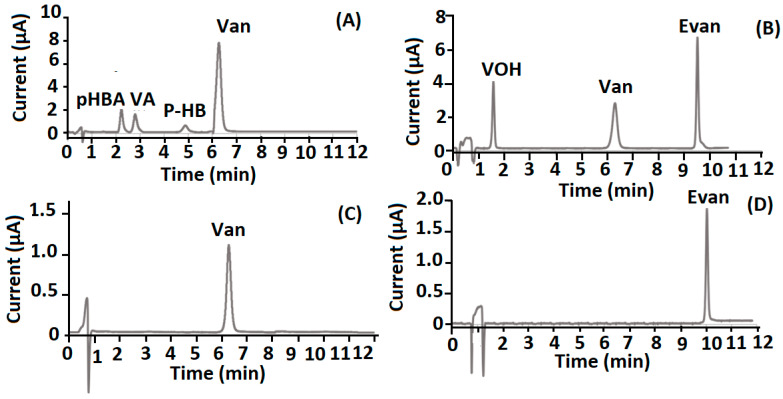
Chromatograms obtained by the analysis of four different selected vanilla extract products (**A**–**D**).

**Table 1 molecules-27-02915-t001:** Figures of merit for the determination of vanilla phenolic compounds by LC-ECD developed method.

Markers	Linear Range (µg g^−1^)	A = (a ± S_a_)C + (b ± S_b_)	R^2^	LOD (µg g^−1^)	LOQ (µg g^−1^)	RSD (%)
V-OH	0.5–25	A = (9.76 × 10^−7^ ± 7.68 × 10^−9^ )C + (1.27 × 10^−7^ ± 3.59 × 10^−8^)	0.9997	0.11	0.36	3.94
p-HBA	0.5–25	A = (3.62 × 10^−7^ ± 3.4 × 10^−9^)C − (1.16 × 10^−7^ ± 1.73 × 10^−8^)	0.9997	0.14	0.47	4.32
VA	0.5–25	A = (1.06 × 10^−6^ ± 1.07 × 10^−8^)C − (1.01 × 10^−8^ ± 5.01 × 10^−8^)	0.9995	0.14	0.46	2.89
p-HB	0.5–25	A = (1.63 × 10^−7^ ± 1.07 × 10^−9^)C − (1.88 × 10^−8^ ± 5.46 × 10^−9^)	0.9998	0.10	0.33	4.33
Van	0.5–25	A = (1.31 × 10^−6^ ± 1.37 × 10^−8^)C + (1.15 × 10^−7^ ± 6.39 × 10^−8^)	0.9995	0.14	0.48	4.76
EVan	0.5–25	A = (1.39 × 10^−6^ ± 1.44 × 10^−8^)C + (3.68 × 10^−7^ ± 6.75 × 10^−8^)	0.9995	0.13	0.48	3.54

**Table 2 molecules-27-02915-t002:** Analysis of vanilla real extracts products (A–D) by LC-ECD method.

Markers	Vanilla Natural Extract (A); µg g^−1^		Vanilla Extract Product (B); µg g^−1^		Vanilla Extract Product (C); µg g^−1^		Vanilla Extract Product (D); µg g^−1^	
	ED Detector	DAD Detector	ED Detector	DAD Detector	ED Detector	DAD Detector	ED Detector	DAD Detector
V-OH	ND	ND	31.07 ± 0.32	33.07 ± 1.21	ND	ND	ND	ND
p-HBA	6.67 ± 0.39	7.12 ± 0.5	ND	ND	ND	ND	ND	ND
VA	9.15 ± 0.19	8.84 ± 0.07	ND	ND	ND	ND	ND	ND
p-HB	5.10 ± 0.01	6.05 ± 0.14	ND	ND	ND	ND	ND	ND
Van	98.57 ± 3.61	101.5 ± 1.38	61.62 ± 0.36	59.2 ± 0.81	10.25 ± 0.31	11.06 ± 0.85	ND	ND
EVan	ND	ND	66.02 ± 0.9	68.41 ± 0.22	ND	ND	11.7 ± 0.08	10.2 ± 0.26

**Table 3 molecules-27-02915-t003:** Predicted values for the assessment of the reliability of the methodology based on non-detected markers involved in the fraud confirmation in sample (C).

Ratio Level Limits ^(1)^: Van/p-HBA = 110 − 53		Ratio Level Limits ^(1)^: Van/p-HB = 20 − 10		Ratio Level Limits ^(1):^ Van/VA = 29 − 15	
p-HBA (µg g^−1^)		p-HB (µg g^−1^)		VA (µg g^−1^)	
Expected Concentration ^(2)^	LOQ	Expected Concentration ^(2)^	LOQ	Expected Concentration ^(2)^	LOQ
0.093–0.19	0.47	0.51–1.02	0.33	0.35–0.68	0.46

^(1)^ Bibliographic values obtained for both limits in the ratio: owest and highest limit values in the ratio [5,23,24,25,26]. ^(2)^ Expected values calculated for the lowest and highest limit values.

**Table 4 molecules-27-02915-t004:** Comparison of different electrochemical electrodes for vanillin determination.

Electrode	Electrochemical Detection	Detection Limit	Ref.
Glassy carbon electrode	Linear sweep voltammetry and square-wave voltammetry	16 µM	[27]
PVC/graphite electrode	Amperometric	290 µM	[28]
Al-ZrO_2_-NPs/SPCE	Amperometric	0.6 µM	This work

## Data Availability

The data presented in this study are available in this article.

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
