# Peer review of "Rapid Sample Screening Method for Authenticity Controlling of Vanilla Flavours Using Liquid Chromatography with Electrochemical Detection Using Aluminium-Doped Zirconia Nanoparticles-Modified Electrode"

_molecules, 2022, doi:10.3390/molecules27092915_

Round 1

Reviewer 1 Report

Dear Authors,

The present manuscript entitled " Rapid Sample Screening Method for Authenticity Controlling of Vanilla Flavours Using Liquid Chromatography with Electrochemical Detection using Aluminium Doped Zirconia Nanoparticles-modified Electrode" by Yassine Benmassaoud, Khaled Murtada, Rachid Salghi, Mohammed Zougagh, and Ángel Ríos (molecules-1701528) describes the development a rapid and sensitive method for frauds identification in vanilla flavors based on six phenolic compounds determination. Liquid chromatography with electrochemical detection using a homemade screen-printed carbon electrode modified with aluminum doped zirconia nanoparticles was used in the studies. The proposed method was characterised by good reproducibility, high sensitivity, and good selectivity in determining the vanilla phenolic compounds.

The present article is written correctly and has a good structure; moreover, it has all the necessary parts. The article is interesting from an analytical and food analysis point of view; therefore, it should interest the reader. The paper meets Molecules' requirements, and I recommend the article for publication in Molecules following the common editing stage. My current decision is a minor revision. More specific comments and observations are presented below.

  1. Abstract. Ranges, not single values, should appear when specifying RSD, LOD, and LOQ.
  2. RSD expressed as a percentage is the coefficient of variation (CV).
  3. Page 3, lines 93-105. Were experimental planning methods used?
  4. Figures. The axes, along with the markers, should be clearer and more visible. Subfigures are sometimes described as A, B ... and sometimes as a, b ... . Please, unify it.
  5. Figures 1 and S1. With such a presentation, the values from the signal axis should be removed, and a scale bar should be added.
  6. For some drawings, it is necessary to remove the border (Figures 2 and 4).
  7. How were the LOD values calculated?
  8. Page 5, line 162. All samples are real, and maybe natural is better?
  9. Please add a section where the natural samples, preparation, and extraction methods will be described in detail.
  10. Page 5, line 169. EC or ECD?
  11. Section 3. Please add countries of origin in addition to companies. What were the parameters of the purified water used? Was a precolumn used?
  12. Has the interference been studied? What can be done in the event of strong interference effects? How would you deal with them? What types of interference effects could occur?

I hope that the comments presented will help improve the article.

Author Response

The present manuscript entitled " Rapid Sample Screening Method for Authenticity Controlling of Vanilla Flavours Using Liquid Chromatography with Electrochemical Detection using Aluminium Doped Zirconia Nanoparticles-modified Electrode" by Yassine Benmassaoud, Khaled Murtada, Rachid Salghi, Mohammed Zougagh, and Ángel Ríos (molecules-1701528) describes the development a rapid and sensitive method for frauds identification in vanilla flavors based on six phenolic compounds determination. Liquid chromatography with electrochemical detection using a homemade screen-printed carbon electrode modified with aluminum doped zirconia nanoparticles was used in the studies. The proposed method was characterised by good reproducibility, high sensitivity, and good selectivity in determining the vanilla phenolic compounds.

The present article is written correctly and has a good structure; moreover, it has all the necessary parts. The article is interesting from an analytical and food analysis point of view; therefore, it should interest the reader. The paper meets Molecules' requirements, and I recommend the article for publication in Molecules following the common editing stage. My current decision is a minor revision. More specific comments and observations are presented below.

  1. Abstract. Ranges, not single values, should appear when specifying RSD, LOD, and LOQ.

Ranges of RSD, LOD, and LOQ have been added to the new version of the manuscript.

  1. RSD expressed as a percentage is the coefficient of variation (CV).

% RSD was calculated by dividing the sample standard deviation by sample mean multiplied by 100. IUPAC recommends avoid the use of the term ‘coefficient of variation’.

  1. Page 3, lines 93-105. Were experimental planning methods used?

The optimization of all experimental parameters were carried out by using the univariate method.

  1. Figures. The axes, along with the markers, should be clearer and more visible. Subfigures are sometimes described as A, B ... and sometimes as a, b ... . Please, unify it.

These comments have been taken into account.

  1. Figures 1 and S1. With such a presentation, the values from the signal axis should be removed, and a scale bar should be added.

Signal axis has been removed and a scale bar has been added from Figures 1 and S1

  1. For some drawings, it is necessary to remove the border (Figures 2 and 4).

The border has been removed from Figures 2 and 4.

  1. How were the LOD values calculated?

LOD values were calculated using the standard deviation of linear regression (SDRegression linear).

  1. Page 5, line 162. All samples are real, and maybe natural is better?

Real changed to natural in the new version of the manuscript.

  1. Please add a section where the natural samples, preparation, and extraction methods will be described in detail.

Section about preparation of samples have been added in the new version of the manuscript.

  1. Page 5, line 169. EC or ECD?

EC has been changed to ECD. Thanks to the reviewer. 

  1. Section 3. Please add countries of origin in addition to companies. What were the parameters of the purified water used? Was a precolumn used?

The information has been added to the new version of the manuscript.

  1. Has the interference been studied? What can be done in the event of strong interference effects? How would you deal with them? What types of interference effects could occur?

In this work the interferences haven’t been studied. The influence of some components in the determination of vanillin can be studied for various common interfering substances in the samples to be analyzed, including vanillic acid, vanillic alcohol, p-hydroxybenzaldehyde and p-hydroxybenzoic acid. These substances are known to be present in natural vanilla.

Reviewer 2 Report

The manuscript “Rapid Sample Screening Method for Authenticity Controlling of Vanilla Flavours Using Liquid Chromatography with Electrochemical Detection using Aluminium Doped Zirconia Nanoparticles-modified Electrode” by Benmassaoud et al. reports a rapid and sensitive technique for frauds determination in vanilla flavors was developed. The method comprises separation by liquid chromatography followed by an electrochemical detection using a homemade screen-printed carbon electrode modified with aluminium doped zirconia nanoparticles (Al-ZrO2-NPs/SPCE). I would suggest authors may take a revision before publication. Here are the comments and suggestions:

  1. The corresponding email addresses are different in the main text and supporting information.
  2. The first letter of chemicals can be in the lower case throughout this manuscript.
  3. In Fig. 1(c), where is the CNT-SPCE from?
  4. What are the sources of the real samples?
  5. Can authors summarize and compare the performance of sensors with various modified electrodes.
  6. The ref. #19 should be correct.   

Author Response

The manuscript “Rapid Sample Screening Method for Authenticity Controlling of Vanilla Flavours Using Liquid Chromatography with Electrochemical Detection using Aluminium Doped Zirconia Nanoparticles-modified Electrode” by Benmassaoud et al. reports a rapid and sensitive technique for frauds determination in vanilla flavors was developed. The method comprises separation by liquid chromatography followed by an electrochemical detection using a homemade screen-printed carbon electrode modified with aluminium doped zirconia nanoparticles (Al-ZrO2-NPs/SPCE). I would suggest authors may take a revision before publication. Here are the comments and suggestions:

  1. The corresponding email addresses are different in the main text and supporting information.

Thank to the reviewer. The corresponding author email has been changed in the new version of the manuscript.

  1. The first letter of chemicals can be in the lower case throughout this manuscript.

The first letter of the chemicals has been changed to the lower case.

  1. In Fig. 1(c), where is the CNT-SPCE from?

CNT-SPCE purchased from Dropsens (Asturias, Spain).

  1. What are the sources of the real samples?

The source of the real samples is the local markets (Ciudad Real, Spain). Please see the end of section 3.1.

  1. Can authors summarize and compare the performance of sensors with various modified electrodes.

A table of comparison of different electrochemical electrodes for vanillin determination is added to the new version of the manuscript. Please see table 4.

  1. The ref. #19 should be correct.   

Ref 19 has been corrected.
